# A DNA Finite-State Machine Based on the Programmable Allosteric Strategy of DNAzyme

**DOI:** 10.3390/ijms24043588

**Published:** 2023-02-10

**Authors:** Jun Wang, Xiaokang Zhang, Peijun Shi, Ben Cao, Bin Wang

**Affiliations:** 1Key Laboratory of Advanced Design and Intelligent Computing, Ministry of Education, School of Software Engineering, Dalian University, Dalian 116622, China; 2School of Computer Science and Technology, Dalian University of Technology, Dalian 116024, China

**Keywords:** allosteric DNAzyme, DNA strand displacement, finite-state machine, nanomachine

## Abstract

Living organisms can produce corresponding functions by responding to external and internal stimuli, and this irritability plays a pivotal role in nature. Inspired by such natural temporal responses, the development and design of nanodevices with the ability to process time-related information could facilitate the development of molecular information processing systems. Here, we proposed a DNA finite-state machine that can dynamically respond to sequential stimuli signals. To build this state machine, a programmable allosteric strategy of DNAzyme was developed. This strategy performs the programmable control of DNAzyme conformation using a reconfigurable DNA hairpin. Based on this strategy, we first implemented a finite-state machine with two states. Through the modular design of the strategy, we further realized the finite-state machine with five states. The DNA finite-state machine endows molecular information systems with the ability of reversible logic control and order detection, which can be extended to more complex DNA computing and nanomachines to promote the development of dynamic nanotechnology.

## 1. Introduction

In nature, living systems regulate their internal reactions through response-timed sequential stimuli signals and thus manifest different functions and forms [1]. For instance, cells can differentiate into types with specific functions due to the different activation times and sequences of transcription factors [2]. Recording the time-related information of biological events can help understand the changes of forms in the development and evolution of organisms [3]. It can be seen that perceiving and decoding temporal information is helpful for analyzing history-dependent responses in biology [4]. Inspired by natural time response, the research and design of devices with the time-response characteristics of stimulus signals provide a basic framework for embedding time-based control behaviors into artificial molecular information processing systems [5,6].

The artificial finite-state machine [7] is a device that can perform memory storage and state switching and respond to temporal input. It provides a powerful tool for recording and processing time-related information and has attracted a wide range of attention from researchers. Owing to its high stability, modifiability, and programmability, DNA has promising applications in biosensing [8,9,10], disease diagnosis [11,12,13], molecular computing [14,15,16,17], information storage [18,19,20], nanomachines [21,22], and quantum computing [23,24]. It is one of the ideal materials for constructing artificial state machines. In recent years, researchers have designed a variety of DNA finite-state machines to explore the effects of temporal input signals. For example, a recombinase-based state machine was constructed to achieve state transition in living cells using DNA excision and inversion operations to control and encode states in DNA sequences [25]. By introducing the concept of a clock signal, a finite-state machine that can demonstrate temporal and logic control was developed, and the state machine enabled performing the synchronous operation of parallel systems [26]. In addition, to overcome the limitation of state machines relying on the reaction of free diffusion systems, a finite-state machine based on DNA origami was developed, which used a spatially constrained DNA strand displacement reaction to improve the reaction rate, and it was applied to explore the effect of different input sequences of miRNA on malignancies [27]. Although DNA finite-state machines are developing rapidly, how to achieve dynamic response and reversible switching between the finite states is still a huge challenge.

DNAzyme [28] is a kind of functional nucleic acid with a special sequence. It has the characteristics of easy synthesis, efficient catalysis, and specific recognition, and it has been widely applied in nanodevices [29,30], logic computing [31,32,33,34,35], dynamic reaction networks [36,37,38,39,40,41], and self-assembly [42,43]. In the study of DNAzyme, the programmable regulation of conformation has always been the focus of research. Many design strategies for the allosteric regulation of DNAzyme have been proposed, including pH control [44,45], metal ion regulation [46,47,48], aptamer priming [49], and toehold-mediated DNA strand displacement [50]. These conformational allosteric strategies induced by diversified inputs can dynamically regulate the function of the system, which provides more possibilities for studying the programmable control of finite-state machines.

In this study, we developed a DNA finite-state machine with dynamic response and reversible switching. First, we proposed a programmable allosteric strategy of DNAzyme, which realizes the dynamic regulation of DNAzyme’s conformation through the intramolecular motion of a reconfigurable DNA hairpin induced by toehold-mediated DNA strand displacement. Second, to make the strategy perform optimally, we optimized the experimental conditions, such as the number of bases in the loop domain and stem domain of the DNAzyme structure, the concentration of the input strand, and the reaction temperature. Subsequently, based on this strategy, a finite-state machine with dynamic response and reversible switching was constructed, which implemented reversible transitions between the two states. Finally, through the modular design of the strategy, a DNA finite-state machine with five states was realized. The next state of the state machine is not only determined by the input signal but also by its current state. The state machine can execute programmable regulation through a reconfigurable DNA hairpin to reduce redundant DNA oligonucleotide sequences, thus providing a simple and modular design idea for building complex molecular information systems. In addition, state machines that can dynamically respond to temporal input for reversible state switching are expected to be widely used in nanomachine control.

## 2. Results

### 2.1. Programmable Allosteric Strategy of DNAzyme

To construct a finite-state machine that can dynamically respond to sequential stimuli signals, we first proposed a programmable allosteric strategy of DNAzyme using the design principles of DNA strand displacement and the intramolecular conformational motion of a DNA hairpin, as shown in Figure 1A. The initial state of this strategy contained the substrate RNA1 and complex Sub1. In the basic structure of complex Sub1, the domain f of strand D1 and the domain g of strand Rt1 contained the DNAzyme’s conserved domain sequence, and the domain d was the lower stem after DNAzyme formation. The core part (domains a and b) was an unfolded DNA hairpin hybridized by the inhibitor T1 to form a rigid double helix structure, which forced the DNAzyme’s conserved domain apart. By adding an input strand FT1 that enabled a reaction with inhibitor T1 through toehold-mediated strand displacement to form complex W1, the inactive complex Sub1 was transformed into an active product HM1, while the core part changed from a rigid double helix structure to a flexible single-stranded structure that made the DNA hairpin perform a spontaneous conformational change to shorten the distance of the DNAzyme’s conserved domain. This resulted in the formation of an active DNAzyme, which recognized and cleaved a specific cleavage site (TrAGG) in the substrate RNA1.

First, we verified the feasibility of the programmable allosteric strategy of DNAzyme by analyzing the results of the polyacrylamide gel electrophoresis experiments, as shown in Figure 1B. In the absence of input strand FT1, the complex Sub1 was in an inactive state due to the separation of the DNAzyme’s conserved domain. The substrate RNA1 and complex Sub1 initially coexisted in the solution and could not react with each other. In lane 4, the corresponding two separate bands Sub1 and RNA1 can be observed. When the input strand FT1 was added, it bound to the inhibitor T1 to form a double-stranded structure W1, and the remaining structure spontaneously formed an active DNAzyme-cleaving substrate RNA1. Therefore, in lane 5, we can observe that substrate RNA1 disappeared, and the corresponding product Q1, double-stranded structure W1, and DNAzyme structure HM1 were formed. In addition, we performed fluorescence analysis on the programmable allosteric strategy of DNAzyme, as shown in Figure 1C. A significant increase in fluorescence was observed when the input strand FT1 was added (curve 1). In contrast, no change in fluorescence was observed without the addition of FT1 (curve 2). This is consistent with the experimental results of polyacrylamide gel electrophoresis, indicating that the programmable allosteric strategy of DNAzyme is feasible.

### 2.2. Parameter Optimization of the Strategy

To further optimize the reaction rate of the programmable allosteric strategy of DNAzyme, we performed a parametric analysis of the cleavage rate for the substrate RNA1. First, we explored the effect of the base numbers of domain d on the cleavage rate of the substrate, as shown in Figure 2B. It was observed that DNAzyme activity gradually improved with the increase of the base number, and the cleaving effect was preferred when the base number was 3 or 4 nt. We chose 3 nt as the subsequent experimental parameter because the sequence should avoid redundancy in the complex. We also tested the effect of different base numbers in domain a and domain b on the cleavage rate of the substrate, as shown in Appendix A. The results show that when domain a was 15 nt and domain b was 7 nt, the effect of the cleavage reaction was optimum. Therefore, we finally chose the stem length of 7 nt and the loop length of 15 nt as the subsequent experimental parameters.

After analyzing the rate parameters of DNAzyme’s cleaving of the RNA1 substrate, we examined the effect of different concentrations of inputs and temperature on the programmable allosteric strategy of DNAzyme. As shown in Figure 3A, the reaction rate made no significant difference at a concentration over 1×; therefore, we chose the concentration ratio of 1:1 for the experiment. Next, we investigated the optimal reaction temperature, as shown in Figure 3B. We can observe that the experimental reaction effect was appropriate at 25 °C.

### 2.3. Design of the State Machine Reversible Conversion

Next, based on the above research on the programmable allosteric strategy of DNAzyme, we developed a finite-state machine with dynamic response and reversible switching. In Figure 4A, the numbered circle {Sa, Sc} represents the state, and the arrow represents the transition from the current state to the next state. The letters on the arrow represent the corresponding input. We have also listed the trigger event of the state machine transition. The state machine realized the reversible conversion function from state Sa to Sc by alternately adding the input strand FT1 and the inhibitor T1. In Figure 4B, the process of reversible transition of the state machine is described in detail. When the state machine is in the Sa state, the reconfigurable DNA hairpin is unfolded by the inhibitor T1; therefore, the complex Sub1 loses the ability to cleave the substrate RNA1, and the three-stranded complex Sub1 and substrate RNA1 coexist in the solution. When the input strand FT1 was present, it reacted with the inhibitor T1, which caused a hairpin conformational change in the complex Sub1 without T1 and brought the conserved domain of the DNAzyme closer. The DNAzyme structure formed by the conformational change could cleave the substrate to release fluorescence signals, indicating a state transition from Sa to Sc. The strand T1 hybridized with the loop portion of the DNA hairpin with the addition of T1, the rigid linear double helix structure forced the DNAzyme’s conserved domain to separate, and the state was changed from Sc back to Sa. Therefore, we executed the sequential response of the temporal signal by alternately adding the input strand FT1 and the inhibitor T1 to realize the function of the dynamic response and reversible switching of the finite-state machine. In order to explore the function of reversible conversion of the state machine, we also verified the reverse inhibition process of the programmable allosteric strategy of DNAzyme, as shown in Appendix A.

To verify the dynamic regulation and reversible transition capabilities of the finite-state machine, we performed a real-time fluorescence experiment. In Figure 4C, the results of both the positive control and negative control experiments are shown. Curve (1) was the positive control experiment, and the curve fluorescence signal rose when only the input strand FT1 was added. In curve (2), complex Sub1 and substrate RNA1 in the solution were incubated for 60 min without adding input strand FT1, and the state machine was in the Sa state. Then, the input strand FT1 was added after 60 min of incubation, and the fluorescence began to show an upward trend, and the reaction speed quickly reached the same level as the positive control. This indicates that the input strand FT1 initiated the activity of DNAzyme, causing the state machine to transition from the Sa state to the Sc state. After another 60 min of reaction, the addition of the inhibitor T1 made the fluorescence signal show a horizontal trend. This indicates that DNAzyme activity was inhibited, and the state machine transitioned to the Sa state again. In curve (3), no change in the fluorescence signal was observed without adding input strand FT1, which indicates that there was no reaction between complex Sub1 and RNA1. We completed the circular transitions from Sa to Sc twice to demonstrate the reversible transition function of the state machine. The fluorescence results show that the reversible state transition design of the state machine is feasible, which can also be verified by the polyacrylamide gel electrophoresis experiment results in Appendix A.

### 2.4. Implementation of a DNA Finite-State Machine with Five States

The output of a finite-state machine is determined not only by the input but also by the current state of the state machine. To prove this feature, we introduced two structural frameworks to increase the number of states of the state machine to demonstrate the scalability and controllability of the state machine. In Figure 5A, the abstract diagram schematically depicts the five states of the state machine. The state machine maintains a state represented by the numbered circles {S0, S1, S2, S3, S4}, and each state contains two morphological structures. The arrows in the diagram represent starting from the current state and ending in the next state. The label on the arrow gives the corresponding input signal. The transition between each state is triggered by adding input signals, and different temporal inputs of the same signal show state transitions in different paths of the state machine. The solid black line represents the actual reaction process, and the pink dotted line represents the unexpected reaction process. In detail, for the same input signal FT1, the S0 state can only be converted to the S1 state but not to the S4 state. The S4 state can only be converted by adding input FT1 in the S2 state. This proves that the state transition of the state machine is determined by both the input signal and the current state, thus demonstrating the memory storage function of the state machine for temporal information.

Figure 5B shows in detail the morphological structures contained in each state and the transition process. When the state machine was in the S0 state, it contained complex Sub1 and complex Sub2. The difference between complex Sub1 and complex Sub2 was that the DNA hairpin base sequences with molecular conformational changes were complementarily paired (labeled with a and a* in the figure). The strand FT1 underwent a strand displacement reaction from the toehold domain of the complex Sub1 when the strand FT1 was added, thus transitioning to the S1 state containing HM1 and Sub2. Notably, the design highlight of this module is that it provides two functional domains for the input strand to perform dual roles, where the input strand can not only activate DNAzyme activity by the strand displacement reaction from the toehold domain but also inhibit another DNAzyme activity by a hybridization chain reaction from the loop region of the DNA hairpin. Then, when FT2 was added, the strand FT2 could perform a hybridization chain reaction from its domain a* with the domain a of the HM1 structure to inhibit DNAzyme HM1 activity and perform a strand displacement reaction from the toehold domain of complex Sub2 to activate DNAzyme HM2. This resulted in a transition from the S1 state to the S3 state containing Com1 and HM2. HM1 was activated again with the addition of T2, which made the state transform back to the S1 state. Finally, the state machine is returned to its original S0 state by adding T1. A linear loop transition from the S0 state to the S2 state to the S4 state and finally back to the initial state could be obtained by the exchanged order of FT1 and FT2. This demonstrates that different sequential inputs of temporal signals resulted in different state reaction paths.

To verify the feasibility of the state machine with the five states’ linear loop transition functions, we performed a real-time fluorescence experiment to show that the state machine can respond dynamically according to different temporal inputs. As shown in Figure 6A, we conducted positive control experiments at the same time. Curve 1 represents that only the input strand FT1 was added to the complex Sub1, complex Sub2, substrate RNA1, and substrate RNA2 mixed solution. The FAM fluorescence shows an upward trend, and no change in ROX fluorescence is observed, indicating that only HM1 was activated. Curve 2 represents that only the input strand FT2 was added to the complex Sub1, complex Sub2, substrate RNA1, and substrate RNA2 mixed solution. The ROX fluorescence shows an upward trend, and the FAM fluorescence signal does not increase, indicating that only HM2 was activated. Curve 3 starts with state S0, and neither the FAM fluorescence signal nor the ROX fluorescence signal change. The DNAzyme HM1 was activated with the addition of FT1, but Sub2 remained unchanged. At this time, the FAM fluorescence rose, and the ROX fluorescence remained level, and the state machine was in the S1 state. Then, we added input FT2 after 60 min of reaction. Due to FT2 having a dual role, FT2 could perform the strand displacement reaction from the toehold domain with Sub2 to activate DNAzyme HM2 activity, resulting in an upward trend in the ROX fluorescence. The reaction speed quickly reached the same level as the positive control. FT2 could also perform a hybridization chain reaction from the loop portion of the HM1 that caused HM1 to become Com1, and at this time, the FAM fluorescence tended to be horizontal and did not rise, indicating that the activity of DNAzyme HM1 was inhibited, and the state was S3. Immediately after strand T2 with the same dual effect as FT2 was added, HM1 was activated, HM2 was inhibited, and the state transitioned from S3 to S1 again. Finally, the state returned to the initial state S0 after the addition of strand T1, at which point neither fluorescence was rising. Figure 6B shows the reaction result of adding FT2 and then FT1, and this demonstrates the effect of different chronological information on the state. The experimental results of the fluorescence show that the linear cycle transition of the five states of the state machine is feasible. In order to further illustrate the controllability and extensibility of the state machine, we also implemented the state machine’s four states linear cyclic transformation, as shown in Appendix A.

## 3. Discussion

In the field of molecular nanoscience, the heterogeneity and randomness of the environment make it very important to understand and reveal the time-dependent information of the signal. Therefore, in this paper, we proposed a programmable allosteric strategy of DNAzyme and further realized a state machine with dynamic regulation and reversible switching. In this strategy, a reconfigurable DNA hairpin was used to achieve reversible control of allosteric DNAzyme. Then, this strategy was designed to build a finite-state machine. We analyzed the time-response characteristics, controllability, and expansibility of the finite-state machine by performing different temporal inputs. 

Compared with previously studied state machines [26,27], we introduced DNAzyme’s allosteric strategy in the design of the state machine, which makes the basic unit of the state machine more simple, modular, and integrated and can build a larger time control system in a programmable manner. In this study, the ingenious structural design of the state machine gives the DNA strand the dual function of inhibiting and activating DNAzyme. The state machine based on allosteric strategy achieves state transition by effecting the conformational change of DNAzyme, therefore exhibiting better dynamic response function. What is more, this state machine is able to revert to its original state after the state transition, and the reusable functions laid the foundation for the control of nanomachines. 

Theoretically, our proposed state machine is scalable and compatible, and more basic units can be integrated to continue expanding the number of state machine states. However, with the enlargement of the system, the increased species may make the biochemical signals in the system more prone to crosstalk due to the decrement of the sequence specificity of the DNA strands. Using sophisticated sequence design and system optimization, the similarity of the sequences can be effectively reduced and the robustness of the system can be improved to ensure the stability of the system. In addition, more reliable ways can be explored to reduce the crosstalk between the biochemical signals, such as antibody-mediated DNA strand displacement, rather than only single-stranded DNA molecules for signal transduction. In future work, this programmable DNA finite-state machine with the sequential input of response time will be further studied to assist and control the self-assembly of complex structures [51], which will provide vital programmable control tools for building complex molecular machines. It is also possible to configure suitable binding sequences to design the activator as aptamers [52] or other biological signals [53] for the control of temporal information such as drug release. This will provide a powerful tool for the construction of intelligent nanomachines.

## 4. Materials and Methods

### 4.1. Materials

The DNA strands used in the experiment were synthesized by Sangon Bio-tech Co., Ltd. (Shanghai, China). The DNA strands without modification were purified via polyacrylamide gel electrophoresis. The substrate strands containing RNA base and fluorophore were purified using high-performance liquid chromatography. The sequences of all strands in the experiment are listed in Appendix A, which were simulated using NUPACK [54] to reduce crosstalk between unrelated domains. All DNA samples were diluted in 1×TE buffer (10 mM Tris·HCl, 1 mM EDTA 2Na, and 12.5 mM MgCl_2_·6H_2_O; pH 8.0) as the stock solution. Nanodrop 2000 (Thermo Fisher Scientific Inc., Waltham, MA, USA) was used to measure the concentration of all DNA strands at an absorbance of λ = 260 nm. The concentration was calculated using the molar extinction coefficient provided by Sangon Bio-tech Co., Ltd.

### 4.2. Assembly Procedure

The corresponding DNA strands were added to the 1× TE/Mg^2+^ buffer solution (10 mM Tris·HCl, 1 mM EDTA 2Na, and 12.5 mM MgCl_2_·6H_2_O; pH 8.0). The samples, after mixing, were annealed in a polymerase chain reaction (PCR) thermal cycler. The annealing assembly procedure was set at 95 °C for 10 min, then cooled at a rate of 1 °C per minute for 70 min, and finally dropped to a constant temperature of 25 °C.

### 4.3. Nondenaturing Polyacrylamide Gel Electrophoresis

Each reaction sample (25 µL, 0.8 µM) was mixed with 60% glycerol solution (3 µL) and run on 10% polyacrylamide gel electrophoresis at a constant voltage of 90 V for 2 h. Gels were stained in Stains-All for 30 min and were imaged using a scanner (CanoScan LiDE 120, Tokyo, Japan).

### 4.4. Fluorescence Experiments

Fluorescence experiments were performed using a real-time PCR system (Bio-Rad, CFX96) equipped with a 96-well fluorescence plate reader. All samples were incubated at 25 °C in the 1× TE/Mg^2+^ buffer. All fluorescence experiments were repeated more than three times to ensure reproducibility.

## Figures and Tables

**Figure 1 ijms-24-03588-f001:**
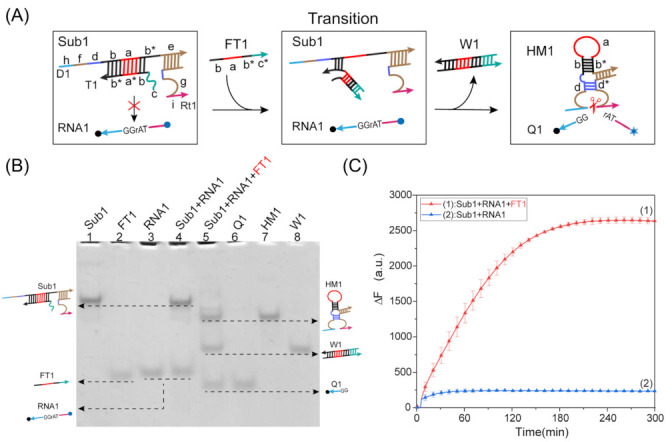
(**A**) Schematic diagram of the programmable allosteric strategy of DNAzyme. The substrate RNA1 was modified by the FAM fluorophore at the 5’ end and the quenching agent BHQ1 at the 3’ end. (**B**) Native PAGE analysis of the programmable allosteric strategy of DNAzyme. ([Sub1] = [RNA1] = [FT1] = 0.8 μM). (**C**) Real-time fluorescence analysis of the programmable allosteric strategy of DNAzyme. Curve (1): the input strand FT1 was added to the mixed solution of complex Sub1 and substrate RNA1. ([Sub1] = [RNA1] = [FT1] = 0.1 μM). Curve (2): only the complex Sub1 and the substrate RNA1 existed in the solution.

**Figure 2 ijms-24-03588-f002:**
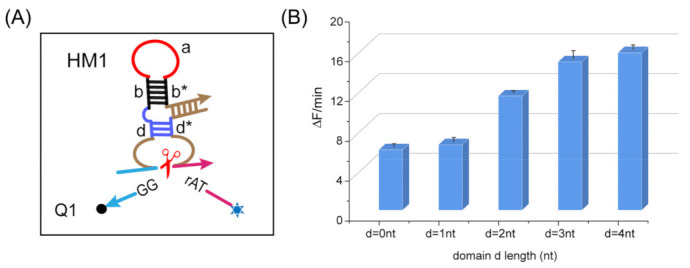
(**A**) Schematic diagram of the DNAzyme-cleaved substrate. (**B**) Comparison of the reaction rate at different base numbers in DNAzyme domain d. Five different base numbers of domain d are listed on the *x*-axis. ([HM1]:[RNA1] = 1:1 = 0.1 μM).

**Figure 3 ijms-24-03588-f003:**
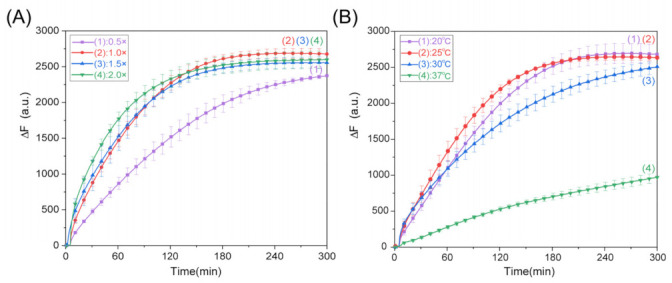
(**A**) Fluorescence analysis for different concentrations of input strand FT1, which were 0.5×, 1.0×, 1.5×, and 2 × 0.1 μM. (**B**) Fluorescence analysis at different reaction temperatures. ([Sub1]:[RNA1]:[FT1] = 1:1:1 = 0.1 μM).

**Figure 4 ijms-24-03588-f004:**
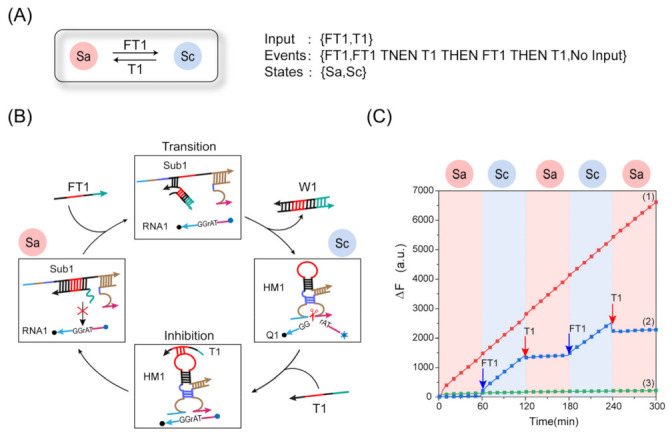
(**A**) Abstract diagram of the reversible transition of the state machine. (**B**) Diagram of the reversible conversion reaction process. (**C**) Fluorescence analysis of the reversible transition of the state machine. Curve (1): the input strand FT1 was added to the mixed solution of complex Sub1 and substrate RNA1. Curve (2): FT1 and T1 were added alternately, [FT1]:[T1]:[FT1]:[T1] = 1:1:2:2, ([Sub1]:[RNA1]:[FT1] = 1:4:1, [Sub1] = [FT1] = 0.1 μM, [RNA1] = 0.4 uM. Curve (3): only substrate Sub1 and substrate strand RNA1 coexisted in the solution.

**Figure 5 ijms-24-03588-f005:**
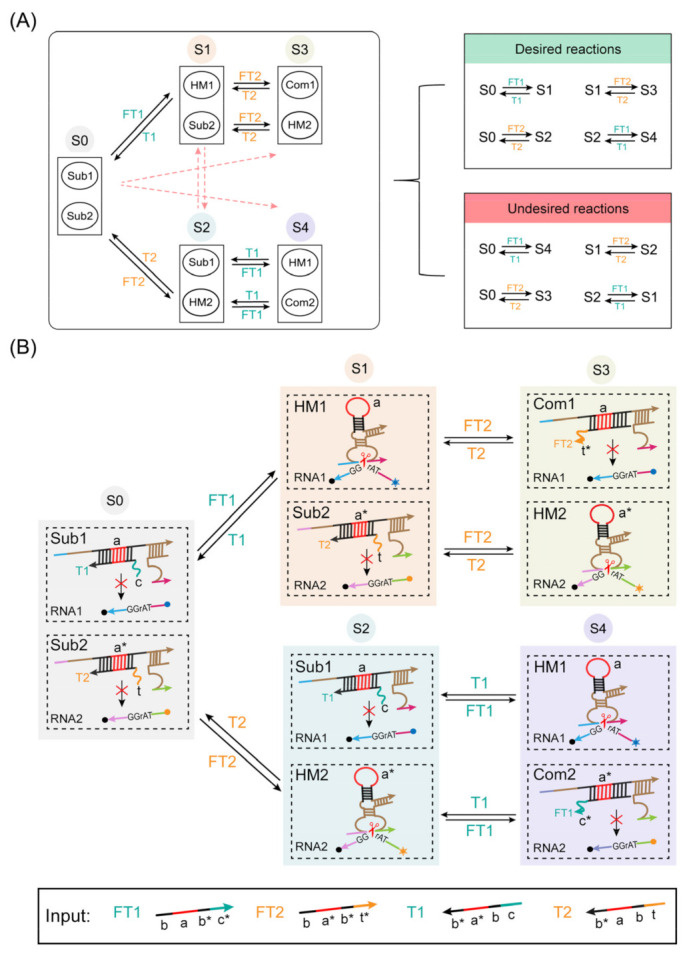
(**A**) Abstract diagram of five state linear loop transitions of the state machine. The solid black line represents the actual reaction process, and the pink line represents the unexpected reaction process. (**B**) Schematic diagram of five state linear loop transitions of the state machine. The blue circle represents FAM fluorescence, and the orange circle represents ROX fluorescence.

**Figure 6 ijms-24-03588-f006:**
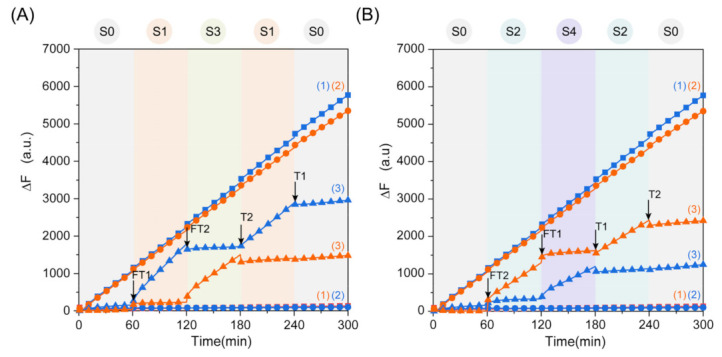
(**A**) Fluorescence analysis of the state cycle transitions when FT1 was added first and then FT2 was added. Curve (1): input strand FT1 was added to the mixed solution of complex Sub1, complex Sub2, substrate RNA1, and substrate RNA2. Curve (2): input strand FT2 was added to the complex Sub1, complex Sub2, substrate RNA1, and substrate RNA2 mixed solution. Curve (3): ([FT1]:[FT2]:[T2]:[T1] = 1:2:2:1, [Sub1]:[Sub2]:[RNA1]:[RNA2] = 1:1:4:4, [RNA1]:[RNA2] = 0.4 μM). (**B**) Fluorescence analysis of the state cycle transitions when FT2 was added first and then FT1 was added. Curve (1): input strand FT1 was added to the mixed solution of complex Sub1, complex Sub2, substrate RNA1, and substrate RNA2. Curve (2): input strand FT2 was added to the complex Sub1, complex Sub2, substrate RNA1, and substrate RNA2 mixed solution. Curve (3): ([FT1]:[FT2]:[T2]:[T1] = 1:2:2:1, [Sub1]:[Sub2]:[RNA1]:[RNA2] = 1:1:4:4, [RNA1]:[RNA2] = 0.4 μM). It was regulated every 60 min.

## Data Availability

Not applicable.

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
