# Peer review of "A DNA Finite-State Machine Based on the Programmable Allosteric Strategy of DNAzyme"

_ijms, 2023, doi:10.3390/ijms24043588_

Round 1

Reviewer 1 Report

The manuscript presents an interesting idea of using DNA segments to make nanodevices capable of processing information. The method described in the manuscript is interesting. What I would though like to know is its shortcomings. I recommend to include a short discussion on that. Beyond that, I also recommend the following modifications:

Line 300: I believe that what you mean by “by changing the conformational change of DNAzyme” should rather be “by effecting the conformational change of DNAzyme”

Line 39-41: I would like to stress that, according to recent research,DNA also finds various applications in quantum computing [1-2]:

  1. Weng-Long Chang, Ju-Chin Chen, Wen-Yu Chung, Chun-Yuan Hsiao, Renata Wong, Athanasios V Vasilakos. Quantum speedup and mathematical solutions of implementing bio-molecular solutions for the independent set problem on IBM quantum computers. IEEE Transactions on NanoBioscience 20(3): 354-376, 2021. DOI: 10.1109/TNB.2021.3075733
  2. Renata Wong and Weng-Long Chang. Fast Quantum Algorithm for Protein Structure Prediction in Hydrophobic-Hydrophilic Model. Journal of Parallel and Distributed Computing 164: 178-190, 2022. DOI: 10.1016/j.jpdc.2022.03.011 

I recommend to accept the manuscript after the suggested minor modifications have been incorporated.

Author Response

Response to Reviewer 1 Comments

Point 1: What I would though like to know is its shortcomings. I recommend to include a short discussion on that.

Response 1: We'd like to thank the reviewers for their careful readings and valuable comments. We have made supplementary explanations in the Discussion part of the manuscript. We have modified the manuscript accordingly. In the Discussion part, we add a description of shortcomings in line 326-334 “However, with the enlargement of the system, the increased species may make the biochemical signals in the system more prone to crosstalk due to decrement of the sequence specificity of the DNA strands. By sophisticated sequence design and system optimization, the similarity of the sequences can be effectively reduced and the robustness of the system can be improved to ensure the stability of the system. Besides, more reliable ways can be explored to reduce the crosstalk between the biochemical signals, such as antibody-mediated DNA strand displacement, rather than only single-stranded DNA molecules for signal transduction”.

Point 2: Line 300: I believe that what you mean by “by changing the conformational change of DNAzyme” should rather be “by effecting the conformational change of DNAzyme”

Response 2: Thanks a lot. We have modified the manuscript accordingly.

Point 3: Line 39-41: I would like to stress that, according to recent research, DNA also finds various applications in quantum computing [1-2]:

Weng-Long Chang, Ju-Chin Chen, Wen-Yu Chung, Chun-Yuan Hsiao, Renata Wong, Athanasios V Vasilakos. Quantum speedup and mathematical solutions of implementing bio-molecular solutions for the independent set problem on IBM quantum computers. IEEE Transactions on NanoBioscience 20(3): 354-376, 2021. DOI: 10.1109/TNB.2021.3075733

Renata Wong and Weng-Long Chang. Fast Quantum Algorithm for Protein Structure Prediction in Hydrophobic-Hydrophilic Model. Journal of Parallel and Distributed Computing 164: 178-190, 2022. DOI: 10.1016/j.jpdc.2022.03.011 

Response 3: We'd like to thank the reviewers for their careful readings and valuable comments. We made corresponding modifications to the manuscript, introduced the field of quantum computing into the description of DNA applications, and added relevant references.

Author Response

Response to Reviewer 2 Comments

Point 1: For the characterization of the real-time fluorescence analysis in figure 1C (including some following plots), I suggest changing the X axis label from cycle to min for clarity.

Response 1: We'd like to thank the reviewers for their careful readings and valuable comments. Indeed, compared with cycles, time can demonstrate the results of the experiment more clearly. We have modified the manuscript accordingly. And the related legend is modified. Meanwhile, for the pictures of supplementary materials, we also changed the X-axis to min.

Point 2: In figure 2B, the author studied the reaction rate at bases from 0 to 4nt. Does it make sense to further investigate length longer than 4nt? Is it going to saturate or decrease?

Response 2: Thank you very much for raising this question. The length of domain d decides the structural stability of the conjugated DNAzyme. If domain d is not long enough, the two parts of the conserved core of the DNAzyme will move apart. Once the structure of the DNAzyme is stable enough, the increment of the length of domain d is no long needed. Figure 2B shows that the reaction rate becomes saturated after the length of domain d is increased to 3nt. When the length of domain d increase to more than 4nt, all the structures of DNAzyme become stable, which means that, the substrates will be digested at a constant rate.

Point 3: Again, in Figure 2B, the error bar is not clear. And to illustrate the reaction rate, is it more accurate to indicate Y axis as delta F/min instead of only delta F?

Response 3: Thank you very much for your valuable questions. It is a very good idea that F/min is used to indicate Y axis, which makes the figure much more intuitive to read. We made corresponding modifications to the Y-axis in Figure 2B, as shown in the attachment.

Point 4: For real world studies, some experiment needs to be performed at 37 degree. Is it possible to include the reaction kinetics at this temperature in figure 3B?

Response 4: Thank you for pointing this out. According to your comments, we added the experimental results of 37℃in the manuscript, as shown in the attachment.

Point 5: Will increase in the length of c promote the strand displacement process and increase the reaction speed?

Response 5: Thank you very much for your valuable comments. As the results shows in reference[1], the DNA strand displacement reaction rate become saturated after the length of the toehold is increased to more than 6nt. Our toehold here is 10nt, so the increase in the length of c will not promote the strand displacement process and the reaction speed is no longer changed.

  • Zhang D Y, Winfree E. Control of DNA strand displacement kinetics using toehold exchange. Journal of the American Chemical Society, 2009, 131(47): 17303-17314.
